# Secondary postpartum hemorrhage: Incidence, etiologies, and clinical courses in the setting of a high cesarean delivery rate

**Natthicha Chainarong, Kittiya Deevongkij, Chusana Petpichetchian**⦾*

Department of Obstetrics and Gynecology, Faculty of Medicine, Prince of Songkla University, Hatyai, Songkhla, Thailand

* chusana020@gmail.com

**Data Availability Statement:** All relevant data are within the manuscript and its Supporting Information files.

## Abstract

### Objectives

To evaluate the incidence, etiologies, and clinical outcomes of secondary postpartum hemorrhage in a hospital with a high cesarean section rate and to compare the etiologies of secondary postpartum hemorrhage following cesarean delivery versus vaginal delivery.

### Materials and methods

This retrospective study included 123 women with secondary postpartum hemorrhage who were treated at a tertiary-level hospital between January 2004 and June 2018. Descriptive statistics and the chi-square test were used for data analysis.

### Results

The incidence of secondary postpartum hemorrhage was 0.21%. The median onset of bleeding was 12 days after delivery. Fifty-two percent of the deliveries were by cesarean section. The most common etiology of secondary postpartum hemorrhage was endometritis (67.5%), followed by retained placental tissue (21.1%). Women who delivered by cesarean section had a higher rate of endometritis (80.0% vs 53.4%) and a lower rate of retained placental tissue (10.8% vs. 32.8%) than those who delivered vaginally. Surgical intervention included uterine evacuation in 29.3% and hysterectomy in 8.1% of the patients. Five percent of women were treated by embolization.

### Conclusions

Endometritis was the most common cause of secondary postpartum hemorrhage. Women who delivered by cesarean section were less likely to have retained placental tissue but were at higher risk for endometritis and uterine pseudoaneurysm than those who delivered vaginally.

**Funding:** The author(s) received no specific funding for this work.

**Competing interests:** The authors have declared that no competing interests exist.

## Introduction

Secondary postpartum hemorrhage (SPPH) is defined as any significant vaginal bleeding that occurs between 24 hours after placental delivery and during the following 6 weeks [1]. The incidence of SPPH has been reported to be 0.2–0.8% [1–3], and it is one of the most common indications for readmission after delivery [4]. Although the number of women affected by SPPH is relatively small compared to the number of women affected by primary postpartum hemorrhage, it can cause serious complications if the diagnosis and appropriate treatments are delayed. One study reported that as many as 22% of patients with SPPH required admission to the intensive care unit [5]. As SPPH usually occurs in the second week after delivery [1, 3, 5], the time at which most patients are discharged from the hospital may contribute to the delay in its detection.

Common causes of SPPH include retention of the placenta, endometritis, and delayed placental bed involution [2, 3]. Other less common etiologies are congenital coagulopathies, cervical cancer, submucous fibroids, placenta adherens, cesarean scar dehiscence, uterine pseudoaneurysm, and uterine rupture [4, 6–8]. Initial management is aimed at achieving hemodynamic stabilization. Subsequently, specific management depends on the cause of bleeding. In most cases, infection is treated in combination with uterine evacuation if retained placental tissue is suspected. In certain circumstances such as continuous bleeding, uterine perforation, and uterine pseudoaneurysm, hysterectomy or arterial embolization may be indicated [1–3, 5].

In previous studies on SPPH, most women included had vaginal deliveries. However, the proportion of women who delivered by cesarean section has been increasing recently. Our hospital is a tertiary-level hospital in a developing country that reported a cesarean delivery rate of 55% in 2016. With cesarean delivery, the chance of bleeding associated with retained placental tissue decreases, while the risks of uterine infection and pseudoaneurysm increase [4]. To date, limited number of studies have reported on SPPH, especially in populations with high cesarean section rates, and none of these studies were conducted in a developing country. Therefore, we conducted this retrospective study with an aim to evaluate the incidence, etiologies, and clinical outcomes of women with SPPH in the setting of a developing country with a high cesarean section rate. The secondary objective was to compare the etiologies of SPPH following cesarean delivery versus vaginal delivery.

## Materials and methods

### Study design and population

This retrospective descriptive study was conducted at a tertiary-level teaching hospital. After the study protocol was approved by the institution's ethics committee, the medical records of all patients evaluated for SPPH who received treatment at the study hospital between January 2004 and June 2018 were reviewed. The need for informed consent was waived because of the retrospective study design.

### Data collection

The extracted information was recorded in a data collection form, which included demographic data, pregnancy-related and other antepartum complications, route of delivery, clinical presentation at the onset of SPPH, treatments, postpartum complications, and etiologies of SPPH. The final diagnosis was ascertained by reviewing the medical record and the radiological, pathological, and microbiological findings when applicable. When more than one diagnosis was given to a patient, the diagnosis that was the most severe or the one that required more specific treatment was chosen as the main diagnosis. The second author was the primary data

collector. To ensure the reliability of the extracted data, the author CP randomly checked 30% of the medical records. Discrepancies were discussed, and if there was a disagreement, an additional opinion from the third author, NC, was used to make a conclusion about the final diagnosis.

## Statistical analysis

The sample size was determined using the formula for estimating the infinite population based on 1) prevalence = 0.8% from a previous study [1], 2) alpha = 0.05, and 3) margin of error (d) = 0.001. The required sample size for an estimation of the prevalence was 30,486.

Descriptive statistics are reported as mean ± standard deviation or median (interquartile range) for continuous variables and number and percentage for categorical variables. The chi-square test was used to compare the etiologies of SPPH between the two delivery groups. R version 3.5.1 (R Core Team) was used to perform statistical analyses. Statistical significance was set at $P < 0.05$.

## Results

There were 46,281 deliveries at the study hospital from January 2004 to June 2018. A total of 123 women were treated for SPPH. Among them, 96 women (78.0%) delivered in the study hospital, and the remaining 27 (22.0%) delivered elsewhere. The incidence of SPPH in women who delivered at the study hospital was one in 482 (0.21%).

Table 1 shows the clinical characteristics of women with SPPH. The majority of them were nulliparous, with a mean age of 30 (17–44) years and median gestational age at delivery of 38 (22–42) weeks. About half of the women had cesarean delivery. The most common indication for cesarean section was prior cesarean section, followed by fetal distress and cephalo-pelvic disproportion.

Among the 123 women who developed SPPH, the most common etiology of SPPH was endometritis in 83 (67.5%) women, followed by retained placental tissue in 26 (21.1%)

**Table 1. Demographic data of women with secondary postpartum hemorrhage (n = 123).**

|  | n (%) |
| --- | --- |
| Age (years)[a] | 30.9 ± 0.5 |
| Parity |  |
| • 0 | 58 (47.2) |
| • 1 | 42 (34.1) |
| • ≥2 | 23 (18.7) |
| Antepartum complications |  |
| • Hypertensive disorders | 10 (8.1) |
| • Diabetes mellitus | 10 (8.1) |
| • Overweight and obesity | 13 (10.6) |
| • Placenta previa | 7 (5.7) |
| • Placenta accreta spectrum | 2 (1.6) |
| Cesarean section in the previous pregnancy | 28 (22.8) |
| Route of delivery |  |
| • Spontaneous vaginal delivery | 43 (35.0) |
| • Operative vaginal delivery | 15 (12.2) |
| • Cesarean section | 65 (52.8) |

[a]Data are presented as mean ± standard deviation.

**Table 2. Etiologies of SPPH in women who delivered vaginally or by cesarean section.**

| Main etiology of SPPH | Route of delivery | | P-value |
|---|---|---|---|
| | Vaginal delivery (n = 58) | Cesarean delivery (n = 65) | |
| Endometritis | 31 (53.4) | 52 (80.0) | <0.01 |
| Retained placental tissue | 19 (32.8) | 7 (10.8) | |
| Other | 8 (13.8) | 6 (9.2) | |
| • Uterine pseudoaneurysm | 0 (0.0) | 4 (6.2) | |
| • Coagulopathy/thrombocytopenia | 3 (5.2) | 0 (0.0) | |
| • Birth canal injury | 4 (6.9) | 0 (0.0) | |
| • Uterine atony | 1 (1.7) | 1 (1.5) | |
| • Cervical cancer | 0 (0.0) | 1 (1.5) | |

Data are presented as n (%).

SPPH, secondary postpartum hemorrhage.

women. The other less common etiologies were uterine artery pseudoaneurysm in four (3.3%) women, birth canal injury in four (3.3%) women, coagulopathy/thrombocytopenia in three (2.4%) women, uterine atony in two (1.6%) women, and cervical cancer in one (0.8%) woman.

Table 2 shows a comparison of the main etiologies of SPPH in women who delivered vaginally and via cesarean section. Endometritis was the most common cause of SPPH in both delivery groups, but it was significantly more common in women who delivered by cesarean section. On the other hand, retained placental tissue was diagnosed in a significantly higher percentage of women who delivered vaginally. Uterine pseudoaneurysm was exclusively diagnosed in women who delivered by cesarean section. Birth canal injury was found only in women who delivered vaginally.

The median time for the onset of bleeding was 12 days after delivery. In 100 (81.3%) women, medical attention was sought after the first episode of bleeding, whereas 20 (16.3%) and three (2.4%) women had two and three bleeding episodes, respectively, before they visited the hospital. At the time of presentation at the hospital, signs of hypovolemic shock were present in 35 (28.5%) women, and fever or pelvic tenderness or both were present in 71 (57.7%).

Table 3 demonstrates the treatment administered to women with SPPH. Medical treatment, including antibiotics with or without a combination of various uterotonic agents, was

**Table 3. Treatments administered to women with secondary postpartum hemorrhage.**

| Treatment | n (%) |
|---|---|
| Antibiotics | 122 (99.2) |
| Uterotonic agents | 49 (39.8) |
| • Oxytocin | 25 (20.3) |
| • Methylergonovine | 40 (32.5) |
| • Misoprostol | 7 (5.7) |
| • Sulprostone | 6 (4.9) |
| Tranexamic acid | 4 (3.3) |
| Uterine evacuation | 36 (29.3) |
| Hysterectomy | 10 (8.1) |
| Other | |
| • Surgical repair | 6 (4.9) |
| • Balloon tamponade | 1 (0.8) |
| • Arterial embolization | 6 (4.9) |

administered to 122 (99.2%) women. Uterine evacuation and hysterectomy were performed in 36 (29.3%) and 10 (8.1%) women, respectively. Among 10 women who required hysterectomy, severe uterine infection unresponsive to conservative treatment was an indication for hysterectomy in seven (5.7%) women. The other indications for hysterectomy were retained placental tissue with intractable bleeding in two (1.6%) women and uterine atony in one (0.8%) woman. Arterial embolization was successfully performed in six (4.9%) women, four (3.3%) of whom were diagnosed with uterine pseudoaneurysm. The other two (1.6%) women who underwent embolization of the uterine artery were initially suspected to have pseudoaneurysm, but computed tomography angiography (CTA) revealed only extravasation of the uterine incision site in one patient and tortuously dilated uterine vessels without evidence of pseudoaneurysm in the other. It was concluded that in these women, the cause of SPPH was endometritis.

Ultrasonography (USG) was performed in 96 (78.0%) women, and in 40 (41.7%) of them, retained placental tissue was suspected. Of 36 (29.3%) women who underwent uterine evacuation, 29 (80.6%) had undergone USG prior to evacuation. In 24 out of 36 (66.7%) women, retained placental tissue was suspected based on pre-evacuation USG and the diagnosis was confirmed by pathological findings in 17 out of 24 (70.8%). In contrast, of the seven (19.4%) women who did not undergo pre-evacuation USG, only two (28.6%) were confirmed to have retained placental tissue.

The most common complication following SPPH was anemia, which was found in 30 (24.4%) women. Serious complications were hemorrhagic shock in 18 (14.6%) women and disseminated intravascular coagulation in three (2.4%) women—one of those three also had liver failure that resolved after supportive treatment. Thirty-six (29.3%) women required blood transfusion. The median number of packed red blood cells (PRBCs) transfused was 2 (1–15) units. Uterine perforation occurred in two women following uterine evacuation. One of these patients was treated conservatively, and the other patient required laparoscopic surgery to stop the bleeding. The duration of hospitalization ranged from 2 to 17 days, with a median duration of 5 days.

## Discussion

The incidence of SPPH in the present study was one in 482 (0.21%), which was similar to that reported in previous studies [1–3]. However, this number may be under-reported because women with minor bleeding may seek medical attention from primary hospitals and not return to our institute. The majority of women in our study were primiparas who delivered at term, similar to those in other studies. The cesarean delivery rate of 55% in our study was much higher than that reported in previous reports (9.0–25.0%) [1–3]. This is possibly because the study hospital is a referral center and many of the women in the study were referred for cesarean section due to various indications.

The median time to onset of bleeding in our study was 12 days, which was similar to that reported in previous studies [1, 3]. The second week after delivery was the most common period during which the bleeding symptom started regardless of the etiology.

The most common etiology of bleeding in our study was endometritis, which accounted for 67.5% of cases, followed by retained placental tissue in 21.1%. This finding was different from that reported in previous studies in which retained placental tissue was the main etiology in 36.7–55.0% of cases [2, 3]. The higher proportion of cesarean delivery in our study can explain these differences because cesarean section is an important risk factor for postpartum endometritis [9, 10] and the chance of retained placental tissue is likely to be reduced by intraoperative exploration of the uterine cavity.

Pseudoaneurysm is a condition in which the arterial wall is injured, followed by the formation of an abnormal arterial flow enclosed within a loose adventitial connective tissue [11].

Uterine pseudoaneurysm can occur after cesarean section and has been regarded as a rare cause of SPPH in numerous case reports [6, 11–13]. Similar to what was reported previously, the four patients with uterine pseudoaneurysm in our study presented with recurrent episodes of bleeding after cesarean delivery; three of them became hemodynamically unstable, and all of them required transfusion of PRBCs ranging from 2 to 7 units. This diagnosis was not found in two older cohort studies with a low cesarean section rate. Only a more recent study [3] found pseudoaneurysm as the etiology of SPPH in 3.3% of women, which is similar to that reported in our study. Although the higher cesarean section rate may explain this finding, improvements in the diagnostic procedures (i.e., the use of Doppler USG and CTA) along with the higher index of suspicion possibly contributed to the increased number of diagnoses.

Subinvolution of the placental bed is diagnosed histologically from a curettage or hysterectomy specimen demonstrating large superficial myometrial vessels with hyaline material replacing the medial layer of the vessel and a partially organized or unorganized endovascular thrombosis [2, 3, 14]. This diagnosis was made and confirmed histologically in 13.3% of patients in a study [3]. Authors of another study in which 55.0% of patients were confirmed to have retained placental tissue hypothesized that the remaining 45.0% may have inadequate involution of the placental site as the cause of SPPH. However, no histological confirmation was made in any of the cases [2]. In our study, 36 curettage and eight hysterectomy specimens did not result in this diagnosis. This was possibly caused, in part, by unawareness of the condition by both the pathologist and gynecologist. Such diagnosis can be made only if the placental site is sampled and selected for a pathological study [14].

Antibiotics were prescribed in 99.2% of the women in our study, similar to previous studies in which antibiotics were administered in 75.0–97.0% of cases [1, 3]. However, uterine evacuation was performed in only 29.3% of the women in our study, whereas it was performed in 50.0–87.6% of those in previous reports [1–3]. Although some authors [1] proposed that uterine curettage has the therapeutic effect of stopping bleeding even though the retained product of conception may not be identified, our study showed that medical treatment alone can be adequate for most women with SPPH. The low rate of uterine evacuation in our cohort might have resulted from the use of USG, which was high (78.0%) compared to that reported in previous studies in which USG was performed in only about one-third of cases [1, 2].

In a more recent cohort in which USG was conducted in 85.0% of cases, the authors found that the use of USG did not affect the rate of surgical intervention [3]. However, our study found that a higher percentage of women were confirmed to have retained placental tissue from the curettage specimen when USG was performed prior to uterine evacuation than when USG was not conducted. In our view, the use of USG is important because it can prevent many women from undergoing unnecessary procedures, especially those who deliver by cesarean section in whom the risk of having retained placental tissue is low and the risk of uterine perforation due to the evacuation procedure might be higher. USG also has an important role in diagnosing uterine pseudoaneurysm in this at-risk group by identification of a hypoechoic mass in the myometrium with low-resistance vascular flow [15].

Uterine artery embolization is regarded as the gold standard of treatment for pseudoaneurysms [16]. In our study, it was successfully performed in all women with uterine pseudoaneurysm. Therefore, the uterus was preserved, and the risk of complications due to hysterectomy was avoided.

In the present study, 10 (8.1%) women required hysterectomy, and seven of them were referred from primary or secondary hospitals after failing conservative treatments. It is understandable that the more severe cases were selectively sent to our institute for definitive surgery, which caused the rate of hysterectomy to be higher than that reported in previous studies (0–5.3%) [1–3].

Among the 16 women who required uterine artery embolization or hysterectomy, 13 (81.3%) delivered by cesarean section, eight (50.0%) had recurrent episodes of bleeding, and nine (56.3%) were hemodynamically unstable at presentation. This information may be useful to classify women who need referral to a tertiary-level hospital where blood components and a multidisciplinary team are available.

The strengths of the present study are that it is the first cohort study to address SPPH in a developing country with a high cesarean section rate, and the sample size was appropriately calculated. Nevertheless, our study has a few limitations. First, as this was a retrospective study, the information retrieved was based on medical records and not all the diagnoses were confirmed by histological or microbiological findings. Second, some women who delivered at our hospital and developed a minor degree of SPPH possibly received treatment elsewhere; therefore, the true incidence of SPPH might be higher than reported herein. Lastly, recruitment bias is possible because women with more severe SPPH were likely to visit or be referred to a tertiary-level hospital. Further prospective study focusing on the occurrence of secondary SPPH, with pre-specified protocol for diagnosis and management of the condition will be of value.

In conclusion, we reported the incidence of SPPH to be 0.21%, with endometritis being the most common etiology, followed by retained placental tissue. Women who delivered by cesarean section were less likely to have retained placental tissue than those who delivered vaginally. Medical treatment was effective in the majority of patients, and only 29.3% required uterine evacuation. USG appeared to have important roles in determining women who would benefit from the uterine evacuation procedure and diagnosing uterine artery pseudoaneurysm, which could be treated successfully with embolization.

## Supporting information

**S1 Data.**
(XLS)

## Acknowledgments

The authors would like to thank Ms. Nannapat Pruphetkaew from the Epidemiology Unit, Faculty of Medicine, Prince of Songkla University for her assistance in data analysis. We also would like to thank Mr. Glenn K. Shingledecker from the International Affair, Faculty of Medicine, Prince of Songkla University for English language editing.

## Author Contributions

**Conceptualization:** Chusana Petpichetchian.

**Data curation:** Natthicha Chainarong, Kittiya Deevongkij, Chusana Petpichetchian.

**Formal analysis:** Kittiya Deevongkij, Chusana Petpichetchian.

**Investigation:** Kittiya Deevongkij, Chusana Petpichetchian.

**Methodology:** Natthicha Chainarong, Kittiya Deevongkij, Chusana Petpichetchian.

**Project administration:** Chusana Petpichetchian.

**Software:** Chusana Petpichetchian.

**Supervision:** Natthicha Chainarong, Chusana Petpichetchian.

**Validation:** Chusana Petpichetchian.

**Writing – original draft:** Kittiya Deevongkij.

**Writing – review & editing:** Natthicha Chainarong, Chusana Petpichetchian.

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
