## [Decision Letter · Decision Letter 0]

14 Feb 2022

Secondary postpartum hemorrhage: Incidence, etiologies, and clinical courses in the setting of a high cesarean delivery rate.

PONE-D-21-34785

Dear Dr. Petpichetchian,

We’re pleased to inform you that your manuscript has been judged scientifically suitable for publication and will be formally accepted for publication once it meets all outstanding technical requirements.

Kind regards,

Federico Ferrari

Academic Editor

PLOS ONE

Additional Editor Comments (optional):

Reviewers' comments:

Reviewer's Responses to Questions

**Comments to the Author**

1. Is the manuscript technically sound, and do the data support the conclusions?

Reviewer #1: Yes

Reviewer #2: Yes

2. Has the statistical analysis been performed appropriately and rigorously? 

Reviewer #1: Yes

Reviewer #2: Yes

3. Have the authors made all data underlying the findings in their manuscript fully available?

Reviewer #1: Yes

Reviewer #2: Yes

4. Is the manuscript presented in an intelligible fashion and written in standard English?

Reviewer #1: Yes

Reviewer #2: Yes

5. Review Comments to the Author

Reviewer #1: Dear Respectable Editor

Thank you to give me the opportunity to review thw manuscript entitled (Secondary postpartum hemorrhage: Incidence, etiologies, and clinical courses in the setting of a high cesarean delivery rate).

Which designed to evaluate the incidence, etiologies, and clinical outcomes of secondary postpartum hemorrhage in a hospital with a high cesarean section rate and to compare the etiologies of secondary postpartum hemorrhage following cesarean delivery versus vaginal delivery.

This retrospective study included 123 women with secondary postpartum hemorrhage who were treated at a tertiary-level hospital.

The study found that the incidence of secondary postpartum hemorrhage was 0.21%. The median onset of bleeding was 12 days after delivery. Fifty-two percent of the deliveries were by cesarean section. The most common etiology of secondary postpartum hemorrhage was endometritis (67.5%), followed by retained placental tissue (21.1%). Women who delivered by cesarean section had a higher rate of endometritis (80.0% vs 53.4%) and a lower rate of retained placental tissue (10.8% vs. 32.8%) than those who delivered vaginally. Surgical intervention included uterine evacuation in 29.3% and hysterectomy in 8.1% of the patients. Five percent of women were treated by embolization.

The study concluded that women who delivered by cesarean section were less likely to have retained placental tissue but were at higher risk for endometritis and uterine pseudoaneurysm than those who delivered vaginally.

The authors provide a structured abstract and informative title

The methodology and the results are clear.

In the discussion they compared their findings, with other authors findings and they mentioned the strength and limitations of their study.

They reached clear conclusions and relevant references.

Finally, I have no suggestions and I think the manuscript discuss an interesting subject to the readers and can be accepted for publication.

Regards

Reviewer #2: The results well represent what is expected from a center with a high cesarean section rate of 52%. The incidence of retained placenta would be low for cesarean delivery when compared to vaginal delivery and also endometritis would be one the highest etiology for secondary postpartum hemorrhage when compared to retained products

Thank you.

6. PLOS authors have the option to publish the peer review history of their article (what does this mean?). If published, this will include your full peer review and any attached files.

Reviewer #1: **Yes: **Ibrahim A. Abdelazim (Professor of Obstetrics and Gynecology, Ain Shams University)

Reviewer #2: No

---

## [Editor Report · Acceptance letter]

21 Feb 2022

PONE-D-21-34785 

Secondary postpartum hemorrhage: Incidence, etiologies, and clinical courses in the setting of a high cesarean delivery rate 

Dear Dr. Petpichetchian:

I'm pleased to inform you that your manuscript has been deemed suitable for publication in PLOS ONE. Congratulations! Your manuscript is now with our production department. 

Kind regards, 

on behalf of

Dr. Federico Ferrari 

Academic Editor

PLOS ONE